# Metabolomics and DNA-Based Authentication of Two Traditional Asian Medicinal and Aromatic Species of *Salvia* subg. *Perovskia*

**DOI:** 10.3390/cells10010112

**Published:** 2021-01-09

**Authors:** Monika Bielecka, Bartosz Pencakowski, Marta Stafiniak, Klemens Jakubowski, Mehdi Rahimmalek, Shima Gharibi, Adam Matkowski, Sylwester Ślusarczyk

**Affiliations:** 1Department of Pharmaceutical Biotechnology, Wroclaw Medical University, Borowska 211A, 50-556 Wroclaw, Poland; bartosz.pencakowski@umed.wroc.pl (B.P.); marta.stafiniak@umed.wroc.pl (M.S.); 2Botanical Garden of Medicinal Plants, Wroclaw Medical University, Jana Kochanowskiego 14, 51-601 Wroclaw, Poland; klemens.jakubowski@umed.wroc.pl (K.J.); mrahimmalek@cc.iut.ac.ir (M.R.); pharmaceutical.biology@wp.eu (A.M.); 3Department of Horticulture, College of Agriculture, Isfahan University of Technology, Isfahan 841583111, Iran; 4Core Research Facility (CRF), Isfahan University of Medical Sciences, Hezar Jerib Street, Isfahan 8174673461, Iran; s.gharibi@mail.mui.ac.ir; 5Department of Pharmaceutical Biology and Botany, Wroclaw Medical University, Borowska 211, 50-556 Wroclaw, Poland; sylwester.slusarczyk@umed.wroc.pl

**Keywords:** phenolic compounds, diterpenoids, tanshinones, *Salvia abrotanoides*, *Salvia yangii*, subgenus *Perovskia*, DNA barcoding

## Abstract

Subgenus *Perovskia* of the extended genus of *Salvia* comprises several Central Asian medicinal and aromatic species, of which *S. yangii* and *S. abrotanoides* are the most widespread. These plants are cultivated in Europe as robust ornamentals, and several cultivars are available. However, their medicinal potential remains underutilized because of limited information about their phytochemical and genetic diversity. Thus, we combined an ultra-high performance liquid chromatography quadrupole time of flight mass spectrometry (UHPLC-QTOF-MS) based metabolomics with DNA barcoding approach based on *trnH*-*psbA* and ITS2 barcodes to clarify the relationships between these two taxa. Metabolomic analysis demonstrated that aerial parts are more similar than roots and none of the major compounds stand out as distinct. Sugiol in *S. yangii* leaves and carnosic acid quinone in *S. abrotanoides* were mostly responsible for their chemical differentiation, whereas in roots the distinction was supported by the presence of five norditerpenoids in *S. yangii* and two flavonoids and one norditerpenoid in *S. abrotanoides*. To verify the metabolomics-based differentiation, we performed DNA authentication that revealed *S. yangii* and *S. abrotanoides* to be very closely related but separate species. We demonstrated that DNA barcoding coupled with parallel LC-MS profiling constitutes a powerful tool in identification of taxonomically close *Salvia* species.

## 1. Introduction

The subgenus *Perovskia* Kar. comprises several species with an ethnobotanical value and medicinal potential. These shrubby, aromatic plants are widespread throughout Central Asia, in an area that includes Iran, Pakistan, Afghanistan, Turkmenistan, Uzbekistan, and Tibet. The ranges of most of the accepted species are limited to the arid montane regions in Central Asia, but two of them, *Salvia abrotanoides* Karel. (previously known as *Perovskia abrotanoides* Kar.) and *Salvia yangii* B.T. Drew (formerly *Perovskia atriplicifolia* Benth.), are distributed over vast areas from Western Iran and Pakistan and as far as Tibet and Xinjiang, in China [1,2,3,4,5,6,7,8,9].

There are a lot of discrepancies regarding the taxonomy of the subgenus *Perovskia*. According to primary classification based on flower morphology, species belonging to *Perovskia* together with four other genera, *Dorystaechas*, *Meriandra*, *Rosmarinus*, and *Zhumeria,* have previously been considered closely related to *Salvia* and were treated together as part of the subtribe Salviinae in the most recent and comprehensive conspectus of Lamiaceae [10].

Later, the field of molecular phylogenetics fundamentally altered the basis of classification of organisms. In 2004, Walker et al., using the *rbcL* and *trnL-F* gene regions and broad sampling within *Salvia*, were the first to reveal that *Salvia* was non-monophyletic [11]. They found *Perovskia*, *Dorystaechas,* and *Rosmarinus*, as well as *Mentha* L., *Origanum* L., and *Thymus* L., and later also *Meriandra* and *Zhumeria* embedded within *Salvia* [11,12].

In 2017, a broad phylogenetic study was undertaken using nuclear markers such as the internal transcribed spacer from nuclear ribosomal DNA (nrITS) and plastid marker *rpl32-trnL* [13]. This study confirmed the polyphyletic character of *Salvia* but also showed the existence of four distinct, well supported evolutionary lineages (Clades I-–IV) within *Salvia* and the inclusion of five additional non-*Salvia* genera, namely, *Dorystaechas*, *Meriandra*, *Perovskia*, *Rosmarinus*, and *Zhumeria*. Based on the plastid data, *Perovskia* and *Rosmarinus* were found in a trichotomy with Clade I (*Salvia* s.s.). The authors reconstructed the evolution and biogeography of *Salvia* and found that the most recent common ancestor of *Perovskia* and *Rosmarinus*, which are sister to *Salvia* Clade I, diverged most likely during the early to middle Miocene. Based on current knowledge, Will and Claßen-Bockhoff propose to split the former large genus *Salvia* into six genera, each supported by geographical distribution, morphology, and karyology, and suggest that they could be elevated to generic rank after revision.

In a related study, low-copy nuclear markers (PPR-AT3G09060, GBSSI) were used to further assess relationships of *Salvia* and related genera within Salviinae [14]. Based upon these results, previous phylogenetic findings, taxonomic, morphological, and practical considerations, the authors concluded that the botanical community would be better served by keeping the genus *Salvia* as traditionally circumscribed, but with the inclusion of 15 species in the five small embedded genera: *Perovskia*, *Dorystaechas*, *Meriandra*, *Rosmarinus*, and *Zhumeria*. Subsequently, to implement this approach and unify all genera within *Salvia*, the authors provided an updated nomenclatural revision, according to which former *Perovskia abrotanoides* is now circumscribed as *Salvia abrotanoides* (Kar.) Sytsma and former *Perovskia atriplicifolia* as *Salvia yangii* B.T. Drew.

*Salvia yangii* B.T. Drew, commonly called Russian sage, is a silver-grey herbaceous perennial half-shrubby plant. All parts of *S. yangii* have silvery hairs, including the flower heads. This plant is native to Central Asia in an area that includes Afghanistan, Eastern Iran, Pakistan, and Tibet [1,15]. Successful over a wide range of climate and soil conditions, *S. yangii* has become popular and is widely planted. Several cultivars/varieties have been developed and cultivated worldwide for ornamental purposes, mainly due to the bountiful blue to violet blossoms arranged in branched panicles.

*Salvia abrotanoides* Kar., called Caspian Russian sage, is also an aromatic erect shrub that occurs in mountains at an elevation of 2200 to 4200 m above sea level from Central and Northeastern Iran across Northern Pakistan to Northwestern India [7].

Both species are closely related and may share very similar morphology, except the leaf shape, which is the main determinative feature, according to many identification keys [1,8,16]. Numerous publications also inform that the characters of the inflorescence, flower, and seeds are similar in both species and they only differ in leaf shape [17,18,19]. The leaf shape is not a reliable determination feature though, as it is very variable within the species, especially in *S. yangii*.

Species of subgenus *Perovskia* range from southwestern to central Asia, with *S. abrotanoides* being more distributed to the west and *S. yangii* to the east. The ranges of the two species overlap in Northeastern Iran, Afghanistan, Pakistan, and Nepal [20]. It is suggested that both species can hybridise in natural habitats [1]. Plants grown from seeds originating from botanical gardens or other plant collections where both species grow in short distance from each other also might be crossed. As both species are able to hybridise, it is not clear which cultivars are variants of *S. yangii* or hybrids between *S. abrotanoides* and *S. yangii* [18,19]. This makes the seemingly simple determination of *Perovskia* species unexpectedly complicated.

Both *S. yangii* and *S. abrotanoides* are used as a folk medicinal herb in the areas of natural occurrence. *S. yangii* is a folk medicinal plant used in Pakistan, Turkmenistan, and Afghanistan [21,22]. The plant has antibacterial activity and is also used as a cooling medicine in the treatment of fever [3,23]. It is known for its flavouring qualities and it is also smoked as a euphoriant and used as a decoction for chronic dysentery [21]. In Uzbekistan and Kyrgyzstan, the aboveground parts are used to heal wounds, and a decoction is used to treat scabies [17]. In traditional Tibetan and Chinese medicine, it has been claimed that *S. yangii* is a powerful analgesic and parasiticide agent [9,24].

*S. abrotanoides*, locally known as brazambol, is used to treat leishmaniasis in Iranian folk medicine. Villagers in the Isfahan Province apply a poultice, made of crushed roots of the plant, water, sesame oil, and wax, on lesions caused by cutaneous leishmaniasis [25,26]. The plant is also used locally for treatment of typhoid, fever, headache, gonorrhoea, vomiting, motion, toothache, cardiovascular diseases, liver fibrosis, painful urination, cough, and as a sedative, analgesic and antiseptic drug [21,27,28,29,30]. Also, the essential oil of *S. abrotanoides* is effective as an antimicrobial agent [31].

Species of subgenus *Perovskia* contain a wide array of natural compounds belonging to different classes [22]. Most of the published data refer to the composition and activities of essential oils from aerial parts [32,33,34,35,36,37,38,39,40,41], indicating a significant variability associated with geographic origin and environmental factors Aerial parts contain also lignans, triterpenes, steroids, and their glycosides, as well as phenolic compounds, including considerable amounts of rosmarinic acid, which is a typical product of the Nepetoideae subfamily of Lamiaceae, known for its numerous health-promoting properties [2,3,7,15,42,43].

Roots of species belonging to the subgenus *Perovskia* are abundant in red-colored quinoid norabietanoids called tanshinones and are considered an alternative source of these valuable pharmacologically active compounds [25,44]. Tanshinones are the main constituents of the well-known Traditional Chinese Medicine herbal drug ‘Danshen’ (*Salvia miltiorrhiza* rhizome et radix a pharmacopoeial drug of China and Europe) [45,46,47,48].

The research performed to date on plants of subgenus *Perovskia* concentrated on either roots [25,44] or aerial parts [7,49,50,51]. There are also reports on phytochemical analysis of the air-dried whole plants [2,3,4,9,24,52]. Plants were collected from their natural habitat or were cultivated. Various methods and solvents were used by researchers to extract natural products of interest which often makes the results hardly comparable. Therefore, we aimed at comprehensive analysis of phytochemical profiles of non-volatile compounds in both roots and leaves of *S. yangii* and *S. abrotanoides* cultivated in Europe under identical climatic and edaphic conditions (Wroclaw, Poland). The ethnopharmacological importance of *S. yangii* and *S. abrotanoides* has prompted us to reanalyse their chemical constituents with potential or proven bioactivity. Here, for the first time, we offer a detailed comparison of phytochemical non-volatile profile in roots and leaves of *S. yangii* and *S. abrotanoides* together with the analysis of a mutual genetic relationship between the two species. The proposed approach of a comparative analysis of closely related plants of a similar but not identical chemical profile may bring tangible results which might potentially pinpoint the outlook for follow-up work through genetic approaches, transcriptomics, proteomics, and other applications. Choosing non-model but potentially important medicinal plants allowed us to test this methodology and prove that the slight differences in morphological and metabolic phenotypes can be verified by molecular analyses, thus facilitating the proper material identification and accelerating the selection of desired traits, breeding, and herb quality control and improvement.

## 2. Materials and Methods

### 2.1. Plant Material

*Salvia abrotanoides* and *S. yangii* plants were grown at the experimental field in the Botanical Garden of Medicinal Plants at the Wroclaw Medical University (Poland). The geographical location of the Garden is as follows: latitude 51°07′03″ N, longitude 17°04′27″ E, altitude 117 m a.s.l. *S. abrotanoides* was introduced from its original habitat in Iranian Karkas Mountains (coordinates: 33°24′57.3″ N 51°46′45.4″ E). Seeds of *S. yangii* were obtained from the collection of the Botanical Garden in Essen (Germany). Cultivars “Lacey Blue” and “Blue Spire” are available commercially as ornamental plants. Plants analyzed in this study were cultivated clonally via vegetative propagation since introduced into the Botanical Garden of Medicinal Plants at the Wroclaw Medical University. Plant voucher specimens were deposited in the herbarium of the Botanical Garden of Medicinal Plants at the Wroclaw Medical University (reference numbers: *S. abrotanoides*: P-122; *S. yangii*: P-123; cultivar “Blue Spire”: P-124; cultivar “Lacey Blue”: P-125).

Leaves and roots of *S. abrotanoides* and wild type *S. yangii* (Appendix A, Figure A1) were collected for phytochemical analysis and were immediately frozen in liquid nitrogen. Harvested leaf samples were finely ground using a mortar and pestle with liquid nitrogen. Root samples were finely ground using an automatic/electric grinder Analysette 3 Spartan and Pulverisette 0 Cryo-Box (Fritsch GmbH, Idar-Oberstein, Germany) filled with liquid nitrogen.

Three accessions of *S. yangii* (including wild type and cultivars “Lacey Blue” and “Blue Spire”) and a single accession of vegetatively propagated *S. abrotanoides* were subject to the molecular study. For phylogenetic analysis, a single, fresh, mature leaf of each plant was harvested and ground to a fine powder in liquid nitrogen.

All samples were stored at −80 °C. The harvest of leaves and roots was done during the flowering season (August 2016).

### 2.2. Sample Preparation for Qualitative Analysis

For direct qualitative analysis, 100 mg of lyophilized plant material was extracted with 5 mL of MeOH: H_2_O (8:2 *v*/*v*) in an ultrasonic bath for 20 min at 25 °C. The supernatant was collected and evaporated to dryness. Extracts were suspended in 2 mL of 0.2% formic acid (FA) in H_2_O prior to preliminary purification and concentration using solid phase extraction (SPE). SPE was performed using an Oasis Max 1cc Vac Cartridge, 500 mg Sorbent per Cartridge, 30 µm Particle Size (Waters Corporation, Milford, MA, USA) conditioned with MeOH (4 mL) and H_2_O (4 mL). The sample was loaded and cartridges washed with 5 mL water in 0.1% FA. The phenolic compounds were eluted with 5 mL of 80% MeOH in 0.1% FA. Samples were re-evaporated to dryness and dissolved in 25% of MeOH to obtain 1 mg dry weight/mL baseline plant material concentration.

### 2.3. Liquid Chromatography–Mass Spectrometry Analysis

Ultra-high performance liquid chromatography quadrupole time of flight mass spectrometry (UHPLC-QTOF-MS) estimation of metabolite contents was carried out on a Thermo Ultimate 3000 RS (Thermo Fischer Scientific, Waltham, MA, USA) chromatographic system coupled to a Bruker Compact (Bruker, Billerica, MA, USA) quadrupole time of flight (QTOF) mass spectrometer, consisting of a binary pump, sample manager, column manager, and the PDA/diode array detector. Separations were performed on a Kinetex C18 column (2.1 × 100 mm, 2.6 μm, Phenomenex, Torrance, CA, USA), with mobile phase A consisting of 0.1% (*v*/*v*) FA in water and mobile phase B consisting of 0.1% (*v*/*v*) FA in acetonitrile. The samples (2 μL) were injected, and a linear gradient from 15% to 70% phase B in phase A was used for 27 min to separate all main compounds. The flow rate was 0.3 mL/min, and the column was held at 30 °C. Mass spectra were acquired in positive-ion mode over a mass range from *m*/*z* 100 to 1500 with 5 Hz frequency. Ultrapure nitrogen was used as drying and nebulizer gas and argon was used as collision gas. The collision energy was set automatically from 15 to 75 eV, depending on the *m*/*z* of the fragmented ion. Operating parameters of the ESI ion source were as follows: capillary voltage 3 kV, dry gas flow 6 L/min, dry gas temperature 200 °C, nebulizer pressure 0.7 bar, collision radio frequency 700.0 V, transfer time 100.0 μs, and pre pulse storage 7.0 μs. Acquired data were calibrated internally with sodium formate introduced to the ion source at the beginning and end of each separation via a 20 μL loop. Processing of spectra was performed with Bruker DataAnalysis 4.3 software. UV DAD data were collected at 190–450 nm, and chromatograms acquired at UV 280 nm were smoothed by use of the Savitzky−Golay algorithm (window width 5 points, one iteration). Then, the detected peaks at retention time and the corresponding MS/MS were integrated. All determinations were done in triplicate (biological replication meaning a different extract *n* = 3); each biological replication was measured at least twice as a technical replication.

ProfileAnalysis software (version 2.1, Bruker Daltonik GmbH, Bremen, Germany) was used to preprocess the raw UHPLC-QTOF-MS data. ProfileAnalysis parameters were set as follows: advanced bucket generation with retention time range of 1.0–24.0 min, mass range of 100–800 *m*/*z*, normalization to sum of peaks, background subtraction, and time alignment; then, analyses were processed with the Find Molecular Futures (FMF) function to create compounds with a signal-to-noise threshold of 3 for peak detection. The generated bucket table consisting of *m*/*z*–retention time pairs and respective compound intensity was exported and uploaded to Metabolists 5.0 (http://www.MetaboAnalyst.ca/) for multivariate statistical analysis. The initial and final retention times were set for data collection. Data were imported into Metabolists 5.0 (http://www.MetaboAnalyst.ca/) online software to estimate missing values and to filter and normalize data (normalization by the median). No transformation was generalized, and the data matrix was mean-centered and divided by the square root of the standard deviation of each variable (Pareto scaling). The obtained data matrix was introduced into SIMCA-P+ 16.01 (Sartorius AG, Göttingen, Germany) software for in-review multivariate statistical analysis of principal component analysis (PCA). The PCA score plot was used to present a natural correlation between the observations. To identify differential compounds, the PLS-DA (Partial Least Squares Discriminant Analysis) model was used to explore differences in depth between the profile metabolome of *Salvia yangii* and *S. abrotanoides* The PLS-DA model with VIP values (VIP ≥ 1.0) and |p(corr)| ≥ 0.5 was used to select differential compounds (PLS-DA data not shown).

### 2.4. Genomic DNA Extraction

DNA was extracted with the modified Doyle & Doyle method [53]. Spectrometric DNA concentration measurements were conducted with a NanoDrop ND 2000C (Thermo Fisher Scientific, Waltham, MA, USA). If needed, DNA isolates were purified with the Syngen DNA clean-up Kit (Syngen Biotech, Wroclaw, Poland) and the DNA concentration was measured again.

### 2.5. DNA Barcoding

ITS rDNA regions were amplified using primers ITS1F [54] and ITS4 [55] while the *trnH*-*psbA* chloroplast intergenic spacer was amplified with *trnH*f_05 [56] and *psbA*3_f [57] primers (Table 1). PCR was performed in a T-100 thermal cycler (BIO-RAD Laboratories, Hercules, CA, USA) in 20 μL volumes containing template DNA and was set up according to the Q5™ High-Fidelity DNA Polymerase manufacturer’s protocol (Q5™ High-Fidelity DNA Polymerase, New England BioLabs Ltd., Hitchin, UK). Thermal cycling conditions were as follows: 95 °C for 30 s; 35 cycles of 95 °C for 10 s; 60 °C (for ITS1F&ITS4) or 62 °C (for *trnH*f_05&*psbA*3_f) for 30 s; 72 °C for 45 s and 72 °C for 2 min for final extension. PCR products were visualized by gel electrophoresis in 2% agarose gel in TEA buffer with 50 bp DNA Ladder (New England Biolabs Ltd., Hitchin, UK) and stained with SimplySafe (EURx, Gdansk, Poland). Amplified DNA products were purified with the Syngen Gel/PCR Mini Kit (Syngen Biotech, Wroclaw, Poland).

Sanger sequencing was carried out with the BrilliantDye™ Terminator v3.1 Kit (Nimagen B.V., Nijmegen, The Netherlands). Reactions were set with a fourfold dilution of the reaction premix and the addition BrilliantDye^®^ Terminator 5X Sequencing Buffer (Nimagen B.V., Nijmegen, The Netherlands) according to the producer’s instructions. Sequencing products were precipitated with ethanol, dissolved in TSR (Hi-Di Formamide) (Thermo Fisher Scientific, Waltham, MA, USA) and then separated by capillary electrophoresis on an Applied Biosystems™ 310 Genetic Analyzer (Thermo Fisher Scientific, Waltham, MA, USA). Two reads were collected for each sample. Eight sequences were generated for molecular and phylogenetic analyses performed along with additional sequences obtained from NCBI (https://www.ncbi.nlm.nih.gov).

Sequence quality was checked with the Sanger Quality Check App (Thermo Fisher Scientific, Waltham, MA, USA). Forward and reverse sequencing reads for each marker region were assembled into a contig using the BioEdit Sequence Alignment Editor [58]. The identity of all obtained sequences was confirmed as ITS or *trnH*-*psbA* through similarity to published sequences using the BLAST algorithm. The ITS2 fragment was extracted with ITSx [59].

Alignments were assembled separately for each marker with the ClustalW algorithm under default conditions in MEGA 7 [60] and were then entered into the program Gblocks v.091b to objectively eliminate poorly aligned positions and divergent regions [61]. Then, a final alignment was assembled by a combination of both alignments calculated for individual markers. The relationship among the taxa was estimated using the Neighbor-Joining (NJ) method [62] and presented as an -rooted phylogram. Due to the large number of gaps in the dataset, a Pairwise deletion Gaps/Missing data treatment was applied. The evolutionary distances were computed using the Maximum Composite Likelihood method [63] according to the general recommendation. To estimate reliability of the tree bootstrap, a 1000-replication test was used [64]. The analysis involved 29 nucleotide sequence combinations including *L. maculatum* and *L. amplexicaule* as an outgroup.

## 3. Results

### 3.1. Metabolic Analysis

Using UHPLC-QTOF-MS analysis, thirty-eight different compounds were found in the methanolic extract of *S. yangii* leaves, while leaves of *S. abrotanoides* comprised thirty-six metabolites extracted with methanol (Table 2; Figure 1). The analysis showed the presence of 14 different phenylpropanoids (3–6, 8–11,14,15,17,18, 21, 27) in leaves of both *S. yangii* and *S. abrotanoides*, including rosmarinic acid (11), considered as one of the chemotaxonomic markers in the Nepetoideae subfamily [22]. In total, 13 diterpenoids were detected in *S. yangii* leaves (20, 22, 23, 28–30, 32, 34, 35, 37, 38, 40, 41) while leaves of *S. abrotanoides* contained 15 different diterpenoids, out of which carnosic acid quinone (31), isorosmanol (33), and trilobinol (39) were found in *S. abrotanoides* and not in *S. yangii*. On the other hand, rosmaridiphenol (35) and sugiol (38) were detected exclusively in *S. yangii* leaves. Among all detected diterpenoids, carnosol (30) turned out to be the most abundant in the leaves of both species as it yielded the highest peak (Figure 1). Both species were found to contain a phenantrene, methoxy-8,11,13-abietatrien-20,11-olide (36), in their leaves. Interestingly, three metabolites identified as triterpenoids (24–26) were detected exclusively in leaves of *S. yangii*. Seven compounds found in leaves of both species were not determined (1, 2, 7, 12, 13, 16, 19).

UHPLC-QTOF-MS analysis of methanolic extracts of *S. yangii* roots revealed the presence of thirty-five different compounds, while in roots of *S. abrotanoides,* only twenty-six metabolites were detected (Table 3; Figure 2). In roots of *S. yangii,* six phenylpropanoids were found (1, 3, 5, 9–11), out of which three, namely, methoxytaxifolin (1), 2-*O*-p-coumaroyltartronic acid (3), and melitric acid B (10), were found only in *S. yangii* and not in *S. abrotanoides*. Roots of *S. abrotanoides* contained four phenylpropanoids, one of which was found exclusively in this species: a flavonoid glucoside–hesperidin (7). Rosmarinic acid (9) and salviaflaside (5), a rosmarinic acid 3′ glucoside, were detected in both species as well as the 8-methylchromen-4-one (12), a phenolic compound belonging to benzopyrans, and salvianolic acid L (6), belonging to phenylnaphthalenes. Chrysoeriol 7-rutinoside (8), which belongs to fatty alcohols/aliphatic alcohols, was detected exclusively in the roots of *S. abrotanoides*. The most abundant group of non-volatile metabolites in roots was diterpenoids, represented by 18 compounds in *S. yangii* and by 13 metabolites in *S. abrotanoides*. Twelve diterpenoids (14, 15, 17–21, 31, 34, 35, 37, 38) were common to both species, including the most abundant, cryptotanshinone (31). Six diterpenoids were unique in the metabolite profile of *S. yangii* roots, namely: didehydrotanshinone IIa (25), acetyloxycryptotanshinone (26), isograndifoliol (27), grandifoliol (28), didehydroacetyloxycryptotanshinone (30), and ketoisogrndifoliol (32). An undetermined cryptotanshinone derivative (29) was found exclusively in *S. abrotanoides* roots. Nine compounds were not determined in methanolic extracts of roots (2, 4, 13, 16, 22, 23, 24, 33, 36) out of which three were detected only in *S. yangii* (2, 24, 36).

### 3.2. Statistical Analysis of UHPLC-QTOF-MS Data

In order to reveal group patterns in non-volatile metabolic profiles of analyzed leaves and roots, an unsupervised PCA was applied. The PC loadings revealed a clear trend to form two clusters among all tested samples, either in leaves or roots, which represent the data in the space formed by the principal components (Figure 3 and Figure 4).

In leaves, the first two principal components (PCs) explained 47.3% and 17.1% of the variation in the spectral data, respectively. On the right side of the score plots (green dots) a strong cluster of *S. abrotanoides* samples formed, while on the left side (blue dots) *S. yangii* samples were well separated (Figure 3a). According to the loadings plot (Figure 3b) these two groups were significantly positively correlated with particular compounds detected in leaves. The PC loadings analyses confirmed that the content of sugiol (38) in *S. yangii* leaves and of carnosic acid quinone (31) in leaves of *S. abrotanoides* were mostly responsible for their chemical differentiation and revealed a clear separation of samples derived from these two species.

In roots, the first principal component explained 65.3% of the variation in the spectral data and the second, 10.2%. Samples of *S. yangii*, represented by blue dots, were closely grouped on the right side of the plot, while on the left side (green dots), *S. abrotanoides* samples occurred (Figure 4a). The loadings plot revealed that these two groups were significantly positively correlated with particular compounds found in roots (Figure 4b). According to the PC loadings, the contents of isograndifoliol (27), acetyloxycryptotanshinone (26), ketoisograndofoliol (32), oxocryptotanshinone (20) and grandifoliol (28) in *S. yangii* roots and the presence of cryptotanshinone derivative (29), chrysoeriol 7-rutinoside (8), and hesperidin (7) in the roots of *S. abrotanoides* were mostly responsible for their chemical differentiation and allowed distinguishing the root samples from the two analyzed *Salvia* species.

### 3.3. Genetic Relationship Analysis

Sanger sequencing resulted in four rDNA ITS2 complete sequences (MT599312, MT599313, MT599314, MT599315) and four *trnH*-*psbA* intergenic spacer sequences (MT815871, MT815872, MT815873, MT815874) obtained for the cultivars S.OBRL.01, S.OBRL.03, S.OBRL.11 and S.OBRL.12, respectively (Table 4). All sequences were subject to BLAST analysis. Due to the lack of publicly available molecular data of the *trnH*-*psbA* barcode in subgenus *Perovskia*, BLAST analysis of the sequenced *trnH*-*psbA* region was not able to indicate similarity to any species from the *Perovskia* subgenus. Instead, the closest homolog of S.OBRL.01 and S.OBRL.11 *trnH*-*psbA* regions indicated by BLAST analysis was the *trnH*-*psbA* sequence of *Salvia miltiorrhiza*, while for S.OBRL.03 and S.OBRL.12, *Salvia californica* and *Salvia chionopeplica* were indicated as the closest homologs, respectively. BLAST analysis of the abundantly available data in the ITS2 region revealed high similarity between *Salvia abrotanoides* and *Salvia yangii* (Table 4).

NJ analysis was performed to estimate the relationship among the studied taxa (*S. yangii*, including wild type and cultivars “Lacey Blue” and “Blue Spire”, and *S. abrotanoides*) in comparison with 23 related *Salvia* species (Table 5). *Lamium maculatum* and *Lamium amplexicaule* were applied as an outgroup. Finally, 50 additional sequences used for the NJ analysis were downloaded from GenBank (25 sequences of ITS2 and 25 sequences of *trnH-psbA*). The manually adjusted alignment of 29 combined ITS2 and 29 *trnH-psbA* sequences was 540 characters long with 361/540 variable sites, 301/540 Parsimony-informative sites.

According to the phylogram reconstructed based on ITS2 and *trnH-psbA* regions, the subgenus *Perovskia* was separated from other sages, yet clearly embedded in *Salvia*, (Figure 5). The topology of the phylogram exhibiteds low interspecies genetic variation between subgenus *Perovskia* representatives indicated by short branches of the tree.

Despite the close genetic relationship, *S. abrotanoides* and *S. yangii* are suggested to be separate species. Moreover, intraspecies genetic variation within the *S. yangii* taxon, between “wild type” *S. yangii* and its cultivars (‘Lacey Blue’ and ‘Blue Spire’), was observed. Furthermore, the intermediate location of *S. yangii* ‘Blue Spire’ between the two *Salvia* species suggests that the ‘Blue Spire’ cultivar might have been created through hybridization between *S. abrotanoides* and *S. yangii* during the cultivation process.

## 4. Discussion

In this work, we have provided a comprehensive analysis of the phytochemical profile of non-volatile compounds separately for roots and leaves of *S. yangii* and *S. abrotanoides* cultivated in Europe (Wroclaw, Poland). A reliable comparison was possible due to ensuring the same cultivation conditions such as: latitude, weather conditions etc., the same sampling method (immediate freezing in liquid nitrogen), and time (middle of the growing season) as well as the same extraction procedure (MeOH: H2O; 8:2 *v*/*v*) and method for chemical analysis (UHPLC-QTOF-MS). Despite ensuring identical cultivation conditions and methodology, a detailed comparison of the phytochemical non-volatile profile in roots and leaves of *S. yangii* and *S. abrotanoides* resulted in detecting several metabolites present in one species and absent in the other. The PC loadings analyses showed that sugiol in *S. yangii* leaves and carnosic acid quinone in leaves of *S. abrotanoides* were mostly responsible for their chemical differentiation. In roots, isograndifoliol, acetyloxycryptotanshinone, ketoisograndofoliol, oxocryptotanshinone, and grandifoliol in *S. yangii* as well as chrysoeriol 7-rutinoside, hesperidin, and one unidentified cryptotanshinone derivative in *S. abrotanoides* allowed for chemical distinguishing of root samples from analyzed species.

As noticed before, most of the published data from aerial parts of species belonging to subgenus *Perovskia* refer to the composition and activities of essential oils, indicating a significant variation [67]. A thorough review of the chemical profile of essential oils from subgenus *Perovskia* is provided by Mohammadhosseini et al. [22]. There are very few reports, however, of non-volatile constituents from aerial parts of species belonging to subgenus *Perovskia*, as a vast majority of them present an analysis of a whole-plant material.

Moreover, the influence of geographic origin and environmental factors influenced the phytochemical profile in both the volatile and non-volatile subset. The extent of such influences in individual cases is hard to assess without extensive experimental research. For example, a light quality modified metabolic profile of cultivated *S. abrotanoides* [68].

Tarawneh et al. [51] isolated four flavoinoids from dried leaves of *S. yangii*, which were not present in our isolates, except for the hydroxy-trimethoxyflavone, detected in leaves of both *S. yangii* and *S. abrotanoides*. Additionally, we detected trihydroxy-dimethoxyflavone, not found by Tarawneh et al. [51].

Amongst ten compounds found in methanolic extracts of air-dried aerial parts of *S. abrotanoides* [7], we detected rosmarinic acid and hesperidin, both present in the leaves of *S. yangii* and *S. abrotanoides*. Interestingly, in our study, hesperidin was also found to be present in the roots of *S. abrotanoides*. Two new tertracyclic diterpens, abrotandiol and abrotanone, found by Khaliq et al. [7] were absent in our extracts. The dihydroxy-methoxymethylflavone, however, found in our leaf isolates, might be the same compound as cismaritin found by Khaliq et al. [7]. Confirming that would require additional NMR analysis.

Amongst diterpenoids found in n-hexane and ethyl acetate extracts of aerial parts of *S. abrotanoides* [69], carnosol, miltirone, rosmanol, and epi-rosmanol were found in methanol leaf extracts of both *S. yangii* and *S. abrotanoides* analyzed here. Miltiodiol, however, has been previously isolated from roots [25,44], which was confirmed by our study. Tabefam et al. [69] also detected ferruginol and dehydroferruginol, both belonging to abietane diterpenoids, which were absent in our isolates. Instead, we detected oxoferruginol (sugiol), present in leaves of *S. yangii*, while leaves of *S. abrotanoides* contained hydroxyferruginol (trilobinol).

Interestingly, our work has shown that the accumulation of triterpenoids, namely, 2-hydroxy-3-oxo-12-oleanen-28-oic acid, 3-oxo-12,18-ursadien-28-oic acid, and trihydroxy-12-ursen-28-oic acid, takes place exclusively in leaves of *S. yangii* and not in *S. abrotanoides*.

In a previous study of *S. yangii* roots (air-dried crude material, extraction with n-hexane), eight diterpenoids were detected [44]. Extraction with 80% methanol performed in this study allowed for the detection of 18 diterpenoids in *S. yangii* roots and 15 in roots of *S. abrotanoides*. 1βOH-cryptotanshinone, cryptotanshinone, tanshinone IIat, and arucadiol (a synonym of miltiodiol) found by Slusarczyk at al. [44] were also detected in our methanolic root extracts in both *S. yangii* and *S.abrotanoides*. Miltirone, however, was present only in the methanolic root extract of *S. yangii* and not in *S. abrotanoides*, which corroborates previous results [25,44]. Two dehydrogenation products, didehydrotanshinone and didehydromiltirone, which were previously detected in n-hexane root extract of *S. yangii*, were not present in our methanolic root extract, neither in *S. yangii* nor in *S. abrotanoides*. Interestingly, isograndifoliol, reported recently in n-hexane root extract of *S. yangii* [66] was also detected in the methanolic root extract of *S. yangii* and was proved to contribute most to chemical differentiation between *S. yangii* and *S. abrotanoides*.

Amongst diterpenoids detected in the EtOAc extracts of native dried roots of *S. abrotanoides* [25], cryptotanshinone, 1βOH-cryptotanshinone, and 1-oxocryptotanshinone were found in methanolic root extracts of both *S. yangii* and *S. abrotanoides*. 1-oxomiltirone, found in both previous studies [25,44], was not detected in this work.

The phytochemistry and chemotaxonomic significance of metabolites found in subgenus *Perovskia* representatives was reviewed by Mohammadhosseini et al. [22].

In this study, we also aimed at analysis of the mutual genetic relationship between *S. abrotanoides* and *S. yangii* rather than their taxonomical position within the *Salvia* subgenera. To our knowledge, no previous work has done that with using DNA barcoding. Usually, a single-entry representative of *Perovskia* subgenus was used for phylogenetic tree reconstructions of *Salvia* [11,12,13,14,70]. There are two reports of employing molecular markers (ISSR markers) to investigate genetic variability among populations from subgenus *Perovskia* growing wild in Iran [71,72]. Heshemifar and Rahimmalek [72] analysed 63 accessions of *S. abrotanoides* from 16 populations and four accessions of *S. yangii*. Thirteen ISSR markers were shown to have a relatively high efficiency in the assessment of population diversity, which was in most cases confirmed by morphological classifications. In the study of Pourhosseini at al. [71], twelve *S. abrotanoides* populations were analyzed and a cultivated population of *S. yangii* was used as an outgroup. The authors have shown that nine ISSR markers are able to genetically distinguish between all studied populations and that the *S. yangii* population was clearly separated from *S. abrotanoides* populations, proving the effectiveness of this DNA fingerprinting method in studying the genetic relationship within the *Perovskia* subgenus. Despite the suitability of the ISSR markers for studying genetic variation among species within subgenus *Perovskia*, we have chosen DNA barcoding as it is regarded as a more sophisticated, standardized, and worldwide acknowledged method, recommended for genetic relationship analysis of land plants [73,74,75].

Although the DNA barcoding has been endorsed as the method of choice for land plant species identification, no consensus has been reached on which plant DNA sequences should be regarded as barcodes. The generally low rate of nucleotide substitution in plant mitochondrial genomes precludes the use of *cytochrome oxidase 1* (*CO1*), a potential equivalent for the standard animal barcode, as a universal plant barcode [76,77]. Instead, the search for a plant barcode has involved looking outside the mitochondrial genome and from the outset, many researchers have accepted that multiple markers will be required to obtain adequate species discrimination [78]. Consequently, the Consortium for the Barcode of Life (CBOL) Plant Working Group has recommended a core-barcode consisting of portions of two plastid-coding regions, *rbcL*+*matK*, to be supplemented with additional markers as required [75]. Limitations of the *rbcL*+*matK* barcodes and idiosyncratic performance of different markers in different taxonomic groups were well discussed by Fazekas et al. [77]. Using additional markers was also strongly suggested by these authors. Beyond the core *rbcL*+*matK* barcode, the most widely used plastid barcoding marker is the intergenic spacer *trnH*-*psbA*. This region is straightforward to amplify across land plants, and is one of the more variable intergenic spacers in plants [79]. Another suggestion is supplementing the phylogenetic analysis of plants with the internal transcribed spacer from nuclear ribosomal DNA (nrITS) [80,81]. A 20% gain in discriminatory power for full ITS and 10–15% gain even when just the ITS2 region is used is an observed and appreciable benefit [82].

Following these recommendations, we have chosen four most widely used and routinely included plant DNA barcodes out of seven leading and 15 known plant sequences considered as barcodes [78]. These were as follows: two plastid coding regions *rbcL* and *matK*, plastid intergenic spacer *trnH*-*psbA,* and nuclear internal transcribed spacer ITS2.

We have amplified, sequenced, and uploaded to GeneBank four *matK* (MT663156, MT663157, MT663158, MT663159) and four *rbcL* (MT663160, MT663161, MT663162, MT663163) fragments obtained for cultivars: S.OBRL.01, S.OBRL.03, S.OBRL.11, and S.OBRL.12, respectively. However, they were not used for phylogenetic analysis as they possessed little (about 1%) genetic diversity within the *Salvia* genus. We have also obtained novel DNA sequences for *trnH*-*psbA* and ITS2 barcodes for our cultivars: S.OBRL.01, S.OBRL.03, S.OBRL.11 and S.OBRL.12, which ensured significantly higher species discrimination success in our study.

In our approach of an extended marker framework for DNA barcodes, a combination of *trnH*-*psbA* and ITS2 barcodes showed sufficient discriminatory power to identify analysed plant organisms to the species level. The phylogram tree constructed with *trnH*-*psbA* and ITS2 barcodes has shown the existence of a separate *Perovskia* clade, which is clearly embedded within *Salvia*, which corroborates the findings of Drew et al. [14], who formally transferred former non-*Salvia* genera, such as *Perovskia*, to *Salvia* and created a subgeneric designation within *Salvia* for each of these former genera.

In the present form, our molecular analysis may suggest that *S. abrotanoides* and *S. yangii* are very closely related but regarded as two separate species. To make this conclusion fully justified, multiple accessions of *S. abrotanoides* would have to be included into the analysis. Unfortunately, at present, we do not have access to other taxonomically undoubtful accessions of *S. abrotanoides;* thus, we decided to rely on what is available. Despite a suggested close genetic relationship, *S. abrotanoides* and *S. yangii* possess distinct phytochemical profiles including qualitative differences within the non-volatile chemicals, which make them a potentially useful model for further analysis. Interestingly, the LC-MS profiles of aerial parts are apparently more similar, and none of the major compounds stand out as distinct, but the PCA processing revealed an obvious separation between the species. Hence, presence or absence of the minor compounds or their varied proportions is also important with such examples as the abovementioned sugiol/trilobinol pair. In contrast, the roots of both species are phytochemically distinct at first sight, and chemometric analysis has verified this fact. Isograndifoliol (Figure 2, peak 27), a hydroxylated norditerpenoid present only in *S. yangii,* is a rarely detected compound that has a high pharmacological potential [66], and its presence in a relatively high amount points out *S. yangii* as an especially valuable medicinal plant. Sugiol, even if a minor compound only, also has significant pharmacological potential, reviewed recently by Bajpai et al. 2020 [83], that can, for example, attenuate STAT3-dependent proliferation and unlocks apoptosis pathways in cancer cells and can block virus entry to cells. Whether or not this compound can contribute to the overall activity of *S. yangii* leaf as a drug has not been explored so far. On the other hand, carnosic acid quinone, a phytochemical marker of *S. abrotanoides* leaves, was reported as a highly potent factor in antioxidant reactions in *S. rosmarinus* and *S. officinalis* [84].

Yet, the phytochemical profiling alone is not sufficient to prove the taxonomic differentiation, especially if easily accessible organs (leaves) would be used for sampling. On the other hand, the results from molecular studies are independent from the sample type and coupled with parallel LC-MS profiling constitute a powerful tool in monitoring of taxonomically close and probably easily crossing species.

Qualitative chemical differences between closely related plant species might be of a different etiology. They might result directly from genetic (and/or epigenetic) differences between species or be an outcome of various processes occurring at the transcriptional and post-transcriptional levels. Comparative analysis of *S. abrotanoides* and *S. yangii* transcriptomes might help discover particular transcripts responsible for detected chemical differences. Performing genetic crosses between *S. abrotanoides* and *S. yangii* would create a segregating population which, through parallel phenotyping and genotyping, might serve for tracing genetic loci potentially responsible for the observed chemical profiles. A similar approach in *Papaver somniferum* allowed for discovering that ten genes from the noscapine biosynthesis pathway are clustered in the poppy genome [85]. In fact, there are numerous reports published to date proving that many secondary metabolites are a product of gene-clusters [86]. Once a certain target genetic locus is found, it could be subject to different biotechnological approaches, such as gene cloning, gene silencing or heterologous expression of proteins, in order to perform a functional analysis of target genes. In the long run, this may lead to developing applied research aiming at optimization of the chemical profile of *Salvia*, which may in turn become an alternative source of bioactive chemicals such as tanshinones.

## Figures and Tables

**Figure 1 cells-10-00112-f001:**
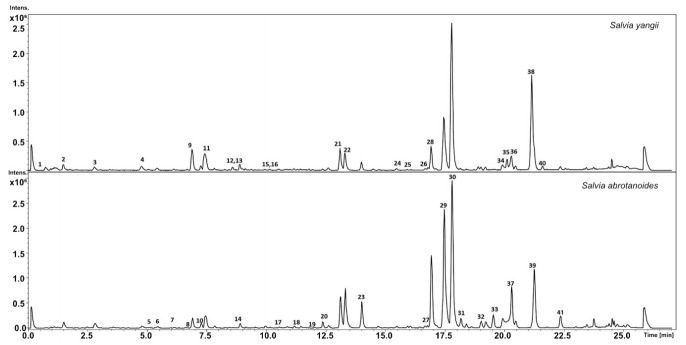
BPC chromatogram of the main metabolites identified in the MeOH extract of leaves of *S. yangii* and *S. abrotanoides*, harvested in the middle of the season.

**Figure 2 cells-10-00112-f002:**
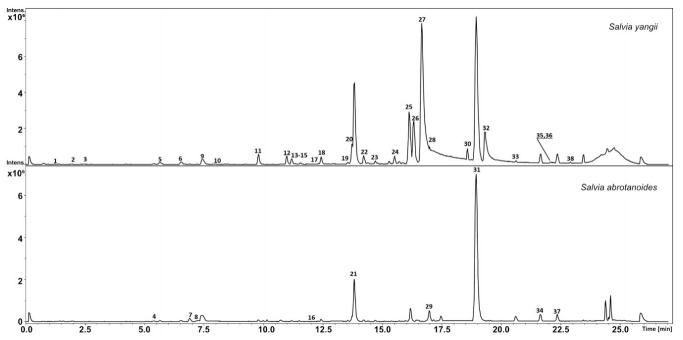
BPC chromatogram of the main metabolites identified in the MeOH extract of roots of *S. yangii* and *S. abrotanoides*, harvested in the middle of the season.

**Figure 3 cells-10-00112-f003:**
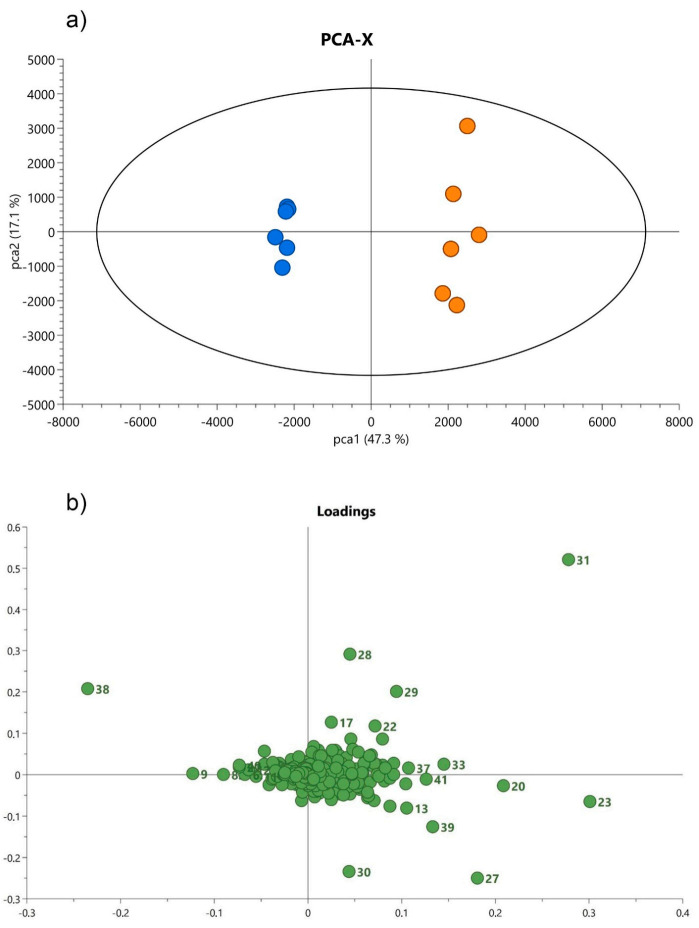
Principal component analysis presenting the relationship between the chemical profile of the main compounds in the leaf and species annotation of the analyzed plants; *S. yangii* (blue) and *S. abrotanoides* (orange) (**a**), with corresponding loadings (PCA model: R^2^X_Cum_ = 0.767, Q^2^_Cum_ = 0.45) (**b**). Each loading number corresponds to the peak number identified in the BPC chromatogram.

**Figure 4 cells-10-00112-f004:**
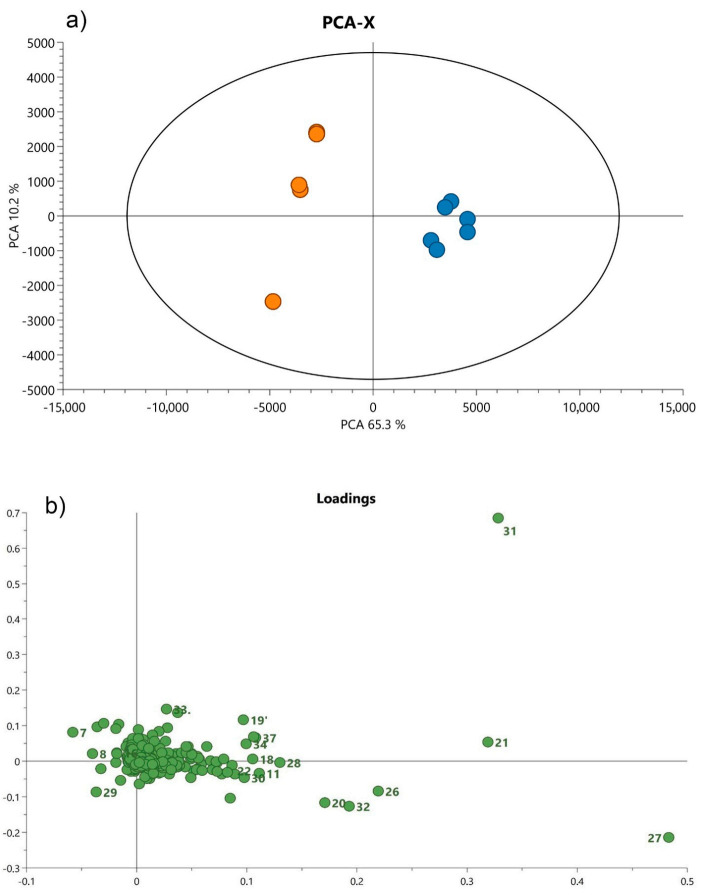
Principal component analysis presenting the relationship between the chemical profile of the main compounds in the root and species annotation of the analyzed plants; *S. yangii* (blue) and *S. abrotanoides* (orange) (**a**), with corresponding loadings (PCA model: R^2^X_Cum_ = 0.755, Q^2^_Cum_ = 0.566) (**b**). Each loading number corresponds to the peak number identified in the BPC chromatogram.

**Figure 5 cells-10-00112-f005:**
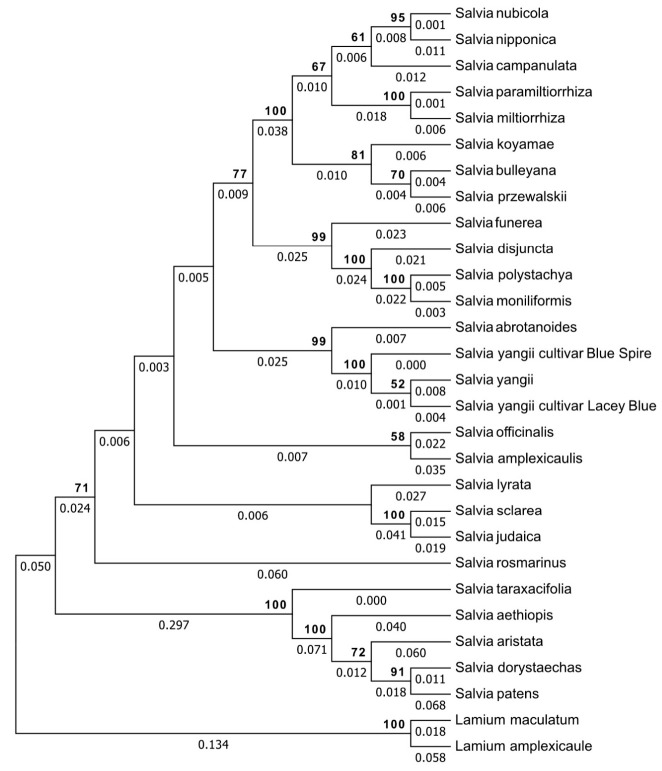
Genetic relationship analysis by the Neighbor-Joining method based on the Maximum Composite Likelihood model and Bootstrap phylogeny test (1000 replicates). Support values are shown above nodes (values lower than 50 are not presented). The branch length is shown below branches. The analysis involved 29 nucleotide sequence combinations. Evolutionary analyses were conducted in MEGA7.

**Table 1 cells-10-00112-t001:** Primers used for amplification and Sanger sequencing of the ITS rDNA region and *trnH*-*psbA* intergenic spacer.

Primer	Sequence (5′ to 3′)	Reference
ITS1F	CTTGGTCATTTAGAGGAAGTAA	[54]
ITS4	TCCTCCGCTTATTGATATGC	[55]
*psbA*3_f	GTTATGCATGAACGTAATGCTC	[57]
*trnH*f_05	CGCGCATGGTGGATTCACAATCC	[56]

**Table 2 cells-10-00112-t002:** Characterization of identified compounds in leaves of *S. yangii* and *S. abrotanoides*, UHPLC–QTOF-MS in positive ion mode.

No.	Compound	RT	UV	*m*/*z*[M + H]^+^	Formula	MS^2^ Main-Ion (Relative Intensity %)	MS^2^ Fragments (Relative Intensity %)	*Salvia yangii*	*Salvia abrotanoides*	References
1	ND	0.9	-	365.1123	C_13_H_16_O_10_	169(100)	271(45), 211(23), 151(18)	***	***	
2	ND	1.7	237, 285, 323	369.1184	C_17_H_20_O_9_	207(100)		***	***	
3	Quercetin 3-rutinoside	2.7	282, 342	611.1609	C_27_H_30_O_16_	303(100)	465(4)	***	***	HMDB0003249
4	Luteolin-O-Hex-Dhex	4.6	260, 343	595.1660	C_27_H_30_O_15_	287(100)	449(3)	***	**	MetFrag
5	Luteolin-O-Pen-Glucuronide	5.1	260, 343	595.1660	C_27_H_30_O_15_	287(100)	395(5), 419(3)	***	***	MetFrag
6	Quercetin 4′-glucoside	5.3	260, 348	465.1025	C_21_H_20_O_12_	303(100)		***	*	HMDB0037932
7	ND	6.1	279, 324	603.2056	C_31_H_38_O_12_	441(100)		***	***	
8	Apigenin-O-Hex-Dhex	6.7	282, 323	579.1706	C_27_H_31_O_14_	271(100)	433(3)	***	**	MetFrag
9	Hesperidin	6.9	284	611.1970	C_27_H_30_O_16_	303(100)	263(5), 449(2), 369(2)	***	*	HMDB0003265
10	Diosmetin-O-Hex-Dhex	7.2	285, 324	609.1815	C_28_H_32_O_15_	301(100)	463(3)	***	***	HMDB0029548
11	Rosmarinic acid	7.4	289, 328	361.0927	C_18_H_16_O_8_	163(100)	181(42), 325(15), 283(10)	***	**	Standard
12	ND	8.6	219, 279	503.1915	C_26_H_30_O_10_	279(100)	341(32), 311(14), 291(25)	***	**	
13	ND	8.8	219, 283, 323	435.1267	C_21_H_22_O_10_	391(100)	349(15)	**	***	
14	Isomargaritene	9.2	-	593.1870	C_28_H_32_O_14_	285(100)	447(12), 429(10)	***	*	HMDB0037415
15	Poncirin (Isosakuranetin-7-O-beta-D-neohesperidoside)	9.2	-	595.2017	C_28_H_34_O_14_	287(100)	269(2)	***	*	HMDB0037487
16	ND	9.3	-	503.1920	C_26_H_30_O_10_	279(100)	311(90), 291(80), 251(70), 259(20)	***	*	
17	Dihydrokaempferide 7-rhamnoside	10.5	282, 320	449.1428	C_22_H_24_O_10_	285(100)		**	***	HMDB0040561
18	Trihydroxy-dimethoxyflavone	11.4	334	331.0813	C_17_H_14_O_7_	270(100)	298(63), 286(4)	***	***	HMDB0128752
19	ND	11.9	-	631.2018	C_31_H_34_O_14_	291(100)	323(65), 259(8)	***	***	
20	Isorosmanol derivative	12.6	-	347.1849	C_20_H_26_O_5_	301(100)	273(15), 241(8), 231(7)	*	***	MetFrag
21	Dihydroxy-methoxymethylflavone	13.1	275, 333	315.0861	C_17_H_14_O_6_	254(100)	282(61), 300(13), 271(7), 285(2)	***	*	MetFrag
22	Rosmanol	13.4	288	347.1858	C_20_H_26_O_5_	301(100)	273(48), 241(27), 283(20)	*	***	MetFrag
23	Epi-rosmanol	14	-	347.1856	C_20_H_26_O_6_	229(100)	273(42), 283(25), 231(18),	*	***	HMDB0035812
24	2-Hydroxy-3-oxo-12-oleanen-28-oic acid	15.4	-	471.3471	C_30_H_46_O_4_	453(100)	425(42), 261(18), 201(12)	***		HMDB0040494
25	3-Oxo-12,18-ursadien-28-oic acid	16	-	453.3377	C_30_H_44_O_3_	407(100)	201(42), 435(10), 205(10)	***		HMDB0037065
26	Trihydroxy-12-ursen-28-oic acid	16.6	-	489.3579	C_30_H_48_O_5_	407(100)	471(25), 435(14)	***		HMDB0036961
27	Hydroxy-trimethoxyflavone	16.8	331	329.1016	C_18_H_16_O_6_	296(100)	329(15), 284(4)	**	***	HMDB0040719
28	11-Methylrosmanol	17	-	361.2020	C_21_H_28_O_4_	329(100)	301(48), 283(15)	*	***	MetFrag
29	7-Methylrosmanol	17.5	-	361.2045	C_21_H_28_O_5_	273(100)	245(90) 301(67), 237(50), 283(41)	***	**	HMDB0035813
30	Carnosol	17.8	284	331.1903	C_20_H_26_O_4_	285(100)	215(45), 267(42), 243(21), 331(20)	***	***	HMDB0002121
31	Carnosic acid quinone	18.2	284	331.1949	C_20_H_26_O_5_	285(100)	303(42), 267(25), 243(18), 225(11), 215(10), 192(8)		***	MetFrag
32	12-Hydroxy-7-oxo-8,11,13-abietatrien-18-al	18.9		315.1938	C_20_H_26_O_3_	287(100)		**	**	HMDB0040746
33	Isorosmanol	19.5	-	347.1858	C_20_H_26_O_5_	301(100)	231(91), 259(85), 283(45), 255(21)		***	HMDB0036661
34	11,12-Dimethylrosmanol	19.9	-	375.2166	C_22_H_30_O_5_	287(100)	329(45), 269(22), 217(15), 191(8)	***	***	HMDB0040525
35	Rosmaridiphenol	20.1	-	317.2104	C_20_H_28_O_3_	181(100)	299(15), 273(14), 261(8)	***		HMDB0037233
36	Methoxy-8,11,13-abietatrien-20,11-olide	20.3	-	329.2112	C_21_H_28_O_3_	217(100)	191(42), 205(33), 231(17), 245(15)	***	***	HMDB0038391
37	Sageone	20.4	-	301.1789	C_19_H_24_O_3_	259(100)	241(27), 272(20), 283(18), 303(13), (216(4)	*	***	HMDB0038684
38	Sugiol	21.1	205, 223, 282	301.2153	C_20_H_28_O_2_	272(100)	191(60), 217(51), 203(42),	***		HMDB0036564
39	Trilobinol	21.2	205, 223, 282	301.2153	C_20_H_28_O_2_	272(100)	191(85), 217(31), 203(22),		***	HMDB0038702
40	Carnosic acid	21.6	-	333.2056	C_20_H_28_O_4_	315(100)	287(65), 273(41), 259(12), 245(5), 233(2)	***	*	Standard
41	Miltirone	22.2	-	283.1678	C_19_H_22_O_2_	225(100)	268(40), 240(28), 253(20)	**	***	[44]

ND: not determined, * ranges from 0.01–0.1 mg/g DM (Dry Mass), ** 0.1–1 mg/g DM and *** above 1 mg/g DM HMDB ID: The Human Metabolome Database MetFrag-MetFrag Online DataBase.

**Table 3 cells-10-00112-t003:** Characterization of identified compounds in roots of *S. yangii* and *S. abrotanoides*, UHPLC–QTOF-MS in positive ion mode.

No.	Compound	RT	UV	*m*/*z*[M + H]^+^	Formula	MS^2^ Main-Ion (Relative Intensity %)	MS^2^ Fragments (Relative Intensity %)	*Salvia yangii*	*Salvia abrotanoides*	References
1	Methoxytaxifolin	1.3	283, 323	335.0763	C_16_H_14_O_8_	271(100)	253(23), 225(14), 197(8)	*		MetFrag
2	ND	1.8	217, 241, 292, 323	549.1594	C_26_H_28_O_13_	531(100)	313(8), 251(4), 295(3)	*		
3	2-O-p-Coumaroyltartronic acid	2.3	-	267.0483	C_12_H_10_O_7_	251(100)	223(23), 206(15)	*		MetFrag
4	ND	5.4	287, 319	511.2535		349(100)	331(47), 258(41), 245(25), 227(17), 211(10)	***	***	
5	Salviaflaside	5.6	288, 319	523.1439	C_24_H_26_O_13_	163(100)	325(45), 287(22), 361(10)	***	***	[65]
6	Salvianolic acid L	6.5	256, 285, 315, 350	719.1599	C_36_H_30_O_16_	521(100)	295(25), 493(18), 221(12), 249(10)	***	***	[65]
7	Hesperidin	6.9	285	611.1982	C_28_H_34_O_15_	303(100)	449(10), 263(4)		***	HMDB03265
8	Chrysoeriol 7-rutinoside	7.3	333	609.1812	C_28_H_32_O_16_	301(100)	463(10)		***	HMDB37453
9	Rosmarinic acid	7.5	288, 319	361.0924	C_18_H_16_O_8_	163(100)	181(42), 325(15), 283(10)	***	***	[44]
10	Melitric acid B	7.9	285, 329	521.1062	C_27_H_20_O_11_	295(100)	493(22), 457(14), 329(6)	***		HMDB0040680
11	3-(3,4-dimethoxyphenyl)-5-hydroxy-7-methoxy-8-methyl-3,4-dihydro-2H-1-benzopyran-4-one	9.8	-	345.1334	C_19_H_20_O_6_	327(100)	283(45), 268(15), 255(8), 201(4)	***	*	HMDB0129573
12	3-(3,4-dimethoxyphenyl)-5-hydroxy-7-methoxy-8-methylchromen-4-one	11	-	343.1170	C_19_H_18_O_6_	325(100)	281(51), 266(18), 251(11), 238(10), 211(4)	***	*	HMDB0129572
13	ND	11.6	218, 339	315.0859	C_17_H_14_O_6_	267(100)	225(15)	*	**	
14	Isograndifoliol derivative	11.8	-	303.1940	C_19_H_26_O_3_	272(100)	203(18), 256(14), 215(7)	***	**	[66]
15	Tanshinone V	12.1	-	315.1210	C_18_H_18_O_5_	297(100)	264(84), 277(21), 236(18)	***	**	[42]
16	ND	12.1		295.1317		267(100)	252(41), 237(28), 225(12)	*	***	
17	Dihydroxycryptotanshinon	12.3	-	329.1381	C_19_H_20_O_5_	265(100)	283(65), 250(18), 311(14), 237(8)	***	*	[44]
18	OH-Tanshindiol A	12.4	-	331.1540	C_19_H_22_O_5_	295(100)	277(42), 267(41), 249(32), 235(21), 225(10)	***	*	[65]
19	OH-cryptotanshinone	13.5	-	313.1435	C_19_H_20_O_4_	267(100)	295(81), 280(74), 249(41), 225(25), 197(20)	***	*	[44]
20	Oxocryptotanshinone	13.7	-	311.1279	C_19_H_18_O_4_	265(100)	283(74), 250(51), 295(14), 237(10), 197(8)	***	*	[44]
21	1-βOH-cryptotanshinone	13.8	220, 267	313.1427	C_19_H_20_O_4_	267(100)	295(81), 280(74), 249(41), 262(25), 197(20), 225(8)	***	*	[44]
22	ND	14.2		329.1388	C_19_H_20_O_5_	311(100)	285(24), 267(15), 252(8), 185(2)	***	*	
23	ND	14.4	-	331.1898	C_19_H_20_O_4_	311(100)	267(14), 283(10), 225(8), 185(5)	***	*	
24	ND	15.5	-	327.1228	C_19_H_18_O_5_	309(100)	265(15), 250(14), 223(11), 281(8)	***		
25	Didehydrotanshinone IIa	16.1	222, 273	293.1163	C_19_H_17_O_3_	278(100)	263(45), 247(41), 232(38), 219(18), 204(5)	***		[44]
26	Acetyloxy cryptotanshinone	16.3	-	355.1548	C_21_H_22_O_5_	295(100)	280(61), 277(48), 267(38), 262(32), 249(15), 247(8)	***		[66]
27	Isograndifoliol	16.7	-	303.1956	C_19_H_26_O_3_	272(100)	203(41), 215(25), 257(21), 187(18), 229(15)	***		[66]
28	Grandifoliol	16.8	-	289.1793	C_18_H_24_O_3_	271(100)	256(45), 241(18)	***		
29	Cryptotanshinone derivative	16.9	-	297.1484	C_19_H_20_O_3_	269(100)	254(21), 239(25), 227(18), 209(3)		***	[42]
30	Didehydro acetyloxy cryptotanshinone	18.6	-	353.1388	C_21_H_20_O_5_	315(100)	293(85), 275(45), 247(28)	***		[65]
31	Cryptotanshinone	18.9	222, 264	297.1469	C_19_H_20_O_3_	269(100)	254(41), 251(38), 279(21), 282(18)	***	*	[44]
32	Ketoisograndifoliol	19.4	-	301.1796	C_19_H_24_O_3_	272(100)	203(45), 215(32), 229(18), 257(12)	***		MetFrag
33	ND	20.4	-	287.1640	C_18_H_23_O_3_	269(100)	254(45), 213(22), 199(18)	***	*	
34	Tanshinone IIa	21.7	-	295.1326	C_19_H_19_O_3_	277(100)	249(71), 262(56), 252(41), 266(20)	***	**	[44]
35	Miltiodiol (Arucadiol)	22	-	299.1636	C_19_H_22_O_3_	271(100)	203(41), 253(24), 183(7)	***	**	[44]
36	ND	22.1	-	269.1520	C_18_H_20_O_2_	254(100)	239(14), 225(12), 215(6)	***		
37	Miltirone	22.2	-	283.1689	C_19_H_22_O_2_	225(100)	268(41), 240(32),253(28)	***	**	[44]
38	6,7-Didehydroferruginol	22.9	-	285.2197	C_20_H_28_O	271(100)	201(12), 272(8)	***	**	

ND: not determined, * ranges from 0.01–0.1 mg/g DM (Dry Mass), ** 0.1–1 mg/g DM and *** above 1 mg/g DM HMDB ID: The Human Metabolome Database MetFrag-MetFrag Online DataBase.

**Table 4 cells-10-00112-t004:** Results of BLAST analysis.

Species (Voucher)	DNA Region	Accession Number (Query)	BLAST Result	Accession Number (Result)	Identity
*Salvia yangii* “Lacey blue” (S.OBRL.01)	ITS2	MT599312	*Perovskia atriplicifolia* (*Salvia yangii*)	KJ584242.1	99.76%
*Salvia yangii* “Lacey blue” (S.OBRL.01)	*trnH*-*psbA*	MT815871	*Salvia miltiorrhiza*	KC473184.1	95.51%
*Salvia yangii* “Blue spire” (S.OBRL.03)	ITS2	MT599313	*Perovskia atriplicifolia* (*Salvia yangii*)	KJ584242.1	99.63%
*Salvia yangii* “Blue spire” (S.OBRL.03)	*trnH*-*psbA*	MT815872	*Salvia californica*	KP852621.1	96.61%
*Salvia yangii* (S.OBRL.11)	ITS2	MT599314	*Salvia yangii*	DQ667223.1	99.62%
*Salvia yangii* (S.OBRL.11)	*trnH*-*psbA*	MT815873	*Salvia miltiorrhiza*	KC473184.1	96.83%
*Salvia abrotanoides* (S.OBRL.12)	ITS2	MT599315	*Salvia yangii*	DQ667223.1	99.64
*Salvia abrotanoides* (S.OBRL.12)	*trnH*-*psbA*	MT815874	*Salvia chionopeplica*	KP852626.1	95.93%

**Table 5 cells-10-00112-t005:** DNA sequences of the ITS2 and *trnH*-*psbA* regions of 23 *Salvia* species used for the NJ analysis. Sequences of *Lamium*
*maculatum* and *Lamium amplexicaule* were used as an outgroup.

No.	Species	Specimen Voucher	ITS2	*trnH*-*psbA*
1	*Salvia aethiopis*	x142 *	DQ667272.1	DQ667370.1
2	*Salvia amplexicaulis*	C756	MG824168.1	MG823900.1
3	*Salvia dorystaechas*	x108 *	DQ667252.1	DQ667360.1
4	*Salvia miltiorrhiza*	SMI4	JQ934135.1	JQ934199.1
5	*Salvia nipponica*	Y. Ibaragi s. n	AB295103.1	MG823977.1
6	*Salvia nubicola*	FLPH 12-124	MG824236.1	MG823979.1
7	*Salvia officinalis*	PS1700MT02	FJ883522.1	FJ513122.1
8	*Salvia paramiltiorrhiza*	SPAR 3	JQ934142.1	JQ934206.1
9	*Salvia przewalskii*	SPRZ 1	JQ934153.1	JQ934217.1
10	*Salvia rosmarinus*	x074 *	DQ667241.1	FJ493283.1
11	*Salvia sclarea*	PS1701MT01	FJ883529.1	FJ513083.1
12	*Salvia koyamae*	n/d	MK425922.1	MG823960.1
13	*Salvia campanulata*	PS1727MT01	FJ883500.1	FJ513135.1
14	*Salvia bulleyana*	Yin et al. 1314	MG824179.1	FJ513128.1
15	*Salvia taraxacifolia*	x001 *	DQ667209.1	DQ667337.1
16	*Salvia judaica*	C769	MG824218.1	MG823957.1
17	*Salvia lyrata*	G.X. Hu QT011	MG824225.1	MG823966.1
18	*Salvia disjuncta*	J. Walker 3018 (WIS)	MF622126.1	MF623945.1
19	*Salvia moniliformis*	Crone 15/9/00 (MJG)	MF622160.1	MF623979.1
20	*Salvia polystachya*	Mexico. Morelos F. Sazatornil 4 (MEXU)	DQ667292.1	MF663893.1
21	*Salvia patens*	x109 *	DQ667253.1	DQ667361.1
22	*Salvia funerea*	JBW 3131	KP852812.1	KP852638.1
23	*Salvia aristata*	x170 *	DQ667280.1	DQ667375.1
24	*Lamium maculatum*	n/d	KF055055.1	HE966679.1
25	*Lamium amplexicaule*	n/d	JX073976.1	HQ966679.1

n/d—not determined * isolate numbers retrieved from Walker and Sytsma, 2007 [12].

## Data Availability

DNA sequence data generated in this study has been deposited in GeneBank at accession numbers: MT599312, MT599313, MT599314, MT599315, MT815871, MT815872, MT815873, MT815874. The UHPLC-QTOF-MS raw data presented in this study are available on request from the corresponding author. The data are not publicly available due to the institutional policy of the authors’ employer.

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
