# Peer review of "Metabolomics and DNA-Based Authentication of Two Traditional Asian Medicinal and Aromatic Species of Salvia subg. Perovskia"

_cells, 2021, doi:10.3390/cells10010112_

Round 1
Reviewer 1 Report
In the manuscript “Metabolomics and DNA-based authentication of two traditional Asian medicinal and aromatic species of Salvia subg. Perovskia”, the authors investigated the taxonomical relation of Salvia yangii and S. abrotanoides by untargeted LC-MS2 metabolomics and sequence of eukaryotic ITS region.
Overall, the manuscript has an interesting subject, i.e., the comparison of chemical composition and genetics between two plants species grown under similar conditions. Still, some comments should be considered before the work is accepted for publication:
- It is not clear the main objective
The authors observed that S. yangii and S. abrotanoides differ from the phenotype and genotype “point of view”. Why are these results relevant? The sentence “Comparative analysis of closely related organisms of a different chemical profile may bring tangible results which might potentially pinpoint the outlook for the follow up work through genetic approaches, transcriptomics, proteomics and other applications.” is too vague. Is there any doubt about the phylogenetic classification of these two species, since genotype analyses have redesigned the phylogeny of the genus Salvia? If that is the case, how this work help to solve the problem.
- Metabolite annotation
How did the authors carry out the metabolic identification? The procedure is not described in the manuscript. If it was done manually, please add all the references.
- Statistical Analysis
Figures 3 and 4 show 6 samples. Which one? How many replicates have been prepared?
How many instrumental replicates were performed? Did the authors prepare any quality control to estimate the coefficient of variation? Authors need to add more details regarding the samples.
The most "significant" compounds in the multivariate analysis are those with the largest peak areas (according to figure 2). This is due to the use of man-centering, which center the data distribution to zero, minimizing collinearity, but do not avoid the “weight” of variables at different units. Shouldn't authors use another methodology, for example auto-scaling, to avoid the impact of the variables' weight?
And what about univariate methods? Why were these analyzes also not incorporated into the study, since they are easily carried out on MetaboAnalyst? The authors would also provide significance values.
- Data availability
The authors should make data available, for example, by adding the MS spectra on the supplementary material, as well as the mass features (m/z_retention time) and their peak areas.
Reviewer 2 Report
The manuscript ‘Metabolomics and DNA-based authentication of two traditional Asian medicinal and aromatic species of Salvia subg. Perovskia’ by Monika Bielecka et al describes a study comprising two approaches, LC-MS based metabolomics combined with PCA analysis and DNA barcoding to analyze distinguishing features of two closely related medicinal plants. Overall, the manuscript is well written, the results are conclusive, and the experimental setup appears to be thoroughly planned. The introduction, however, is a bit lengthy and could be shortened, as well as parts of the discussion, both of which read a bit like a review.
The findings of the metabolomics approach are clearly discussed and well interpreted, and show that indeed, phytochemical profiling reveals distinct compositions between the two Salvia species. Although the phytochemical profile from leaf samples is somewhat similar, PCA analysis reveals a clear clustering according to the two species. For root samples, the differences are more obvious.
Furthermore, the authors used genetic barcoding to investigate the genetically closely related species, and identified differences as well. Here, the discussion of the data is not as well conducted as for the first approach. I am not an expert in the field of phylogenetic analyses and in my opinion, although the literature is amply cited and discussed, the findings of this study would benefit from a clearer presentation and discussion, which would make it easier to follow.
Of note, a strength of the set-up are the controlled conditions of the plant elevation and sampling, which enable a direct comparison of the phytochemical profiles. It would be very interesting, however, to analyze whether seasonal or environmental variations lead to stronger changes in the phytochemical profiles than those observed between two different species. The discussion would benefit by adding information on the dependence of the phytochemical composition and known variations due to climate/region, maybe also as part of the outlook. Furthermore, is the difference found on a molecular level related to the pharmaceutical application of the plant? What is known about the bioactivities of metabolites that are only detected in one of the two plant species, such as sugiol or carnosic acid quinone? A few sentences of the implications could be added to the outlook in regards to the use as medicinal plant.
Please find below my more specific comments (page numbers are unfortunately not consistent!):
Page 5, line 196: please specify ‘biological replicates’: different extract, leaf, or plant?
Page 8, Figure 1 and 2: please use a larger font for axis labeling, the small font is difficult to read.
Page ‘1’, line 341, Table 2: MS2 main ion: there are often more than one listed, what is the threshold to characterize them as main? Furthermore, please explain the * to *** columns of the two species in the table caption. I assume this is abundance, what are the ranges? The same applies for Table 3.
Page ‘2’, line 343, Table 3: For some values in the MS2 column, decimal places are indicated for some of the values (e.g. No 4, 7,8), but for most of them not. Please be consistent.
Page ‘1’, Figure 3 and Page ‘2’, Figure 4: I would recommend using different colors for the PC loading (b) and Salvia abrotanoides in (a), both of which are either the same or a different shade of green. This can lead to confusion.
Author Response
Reviewer 2
Dear Reviewer,
Thank you very much for the detailed revision of our manuscript. We truly appreciate your positive feedback. We also do appreciate the time and effort spend to revise our work. We did our best to suitably address all issues raised by performing appropriate changes into the manuscript text and provided explanations in the correspondence below (in blue). We have also made amendments in the Figure 1, 2, 3 and 4 and provided their revised versions.
We do hope that you will find this revised manuscript appropriate for publishing in Cells.
Best regards,
Monika Bielecka & Co-authors
Detailed responses:
The introduction, however, is a bit lengthy and could be shortened, as well as parts of the discussion, both of which read a bit like a review.
Yes, indeed both sections are not quite short. We revised the text for brevity. However, upon suggestions of other Reviewers and the request for more discussion on phytochemical diversity (below), we had to add some information, too. Nonetheless, the overall volume has been reduced, as can be seen in the ‘track changes’ of the MS Word file.
Furthermore, the authors used genetic barcoding to investigate the genetically closely related species, and identified differences as well. Here, the discussion of the data is not as well conducted as for the first approach. I am not an expert in the field of phylogenetic analyses and in my opinion, although the literature is amply cited and discussed, the findings of this study would benefit from a clearer presentation and discussion, which would make it easier to follow.
As requested, we did our best to communicate our findings plainly. We do hope that it is now easier to follow but would welcome any further suggestion on which specific fragments are perhaps too intricate.
It would be very interesting, however, to analyze whether seasonal or environmental variations lead to stronger changes in the phytochemical profiles than those observed between two different species. The discussion would benefit by adding information on the dependence of the phytochemical composition and known variations due to climate/region, maybe also as part of the outlook. Furthermore, is the difference found on a molecular level related to the pharmaceutical application of the plant?
We fully agree with this opinion, in fact, we have already performed qualitative analysis of the phytochemicals in roots and leaves of both species, in order to follow their variations during the vegetative season. This, together with the expression analysis of genes encoding for enzymes from specific biosynthetic pathways, will serve for another publication. Additionally, we are also analysing if and how the variations in metabolome are reflected by the transcriptome, using the comparative NGS approach (which we are going to publish as a third paper).
Current manuscript is planned as a first out of a series of three publications on detailed analysis of S. abrotanoides and S. yangii and their relationships, including the correlation between molecular markers, transcriptomics and composition of pharmacologically active compounds (with focus on the neuroprotective and selectively cytotoxic actions – compare Ślusarczyk et al. 2020, Int J Mol Sci). However, this is clearly beyond the scope of the present study.
We have also discussed such biodiversity aspects, but for sake of brevity, it has not been very elaborate. Also, in these two species, almost nothing is known except the single previous paper on volatile compounds in S. abrotanoides from Iran (Ghaffari et al. 2018, Chem & Biodiv) and this paper, if published, should also stimulate this kind of future research.
What is known about the bioactivities of metabolites that are only detected in one of the two plant species, such as sugiol or carnosic acid quinone? A few sentences of the implications could be added to the outlook in regards to the use as medicinal plant.
The pharmacological implication of sugiol presence have been mentioned, with reference to the very recent review on this compound (Bajpai et al 2020); The postulated involvement of carnosic acid quinone in antioxidant reactions has been also mentioned (according to Masuda et al. 2002).
Page numbers are unfortunately not consistent!
This has happened due to introducing sections into the manuscript file, which were needed to present large tables (Table 2 and 3) in a landscape orientation. The page numeration has been corrected.
Page 5, line 196: please specify ‘biological replicates’: different extract, leaf, or plant?
Biological replicate represents different extract, each prepared from another 100 mg of lyophilized plant material being a pool of about 20-30 leaves or 5-10 roots.
Page 8, Figure 1 and 2: please use a larger font for axis labeling, the small font is difficult to read.
Fonts for X and Y axis labelling have been enlarged, revised versions of Figure 1 and 2 are provided.
Page ‘1’, line 341, Table 2: MS2 main ion: there are often more than one listed, what is the threshold to characterize them as main?
Thank you for this comment. This has been corrected, the main ion with relative intensity percent is now presented in this column, in Table 2 and 3.
Furthermore, please explain the * to *** columns of the two species in the table caption. I assume this is abundance, what are the ranges? The same applies for Table 3.
Yes, indeed, the asterisks in Table 2 and 3 represent relative abundance of particular chemical. The ranges are:
* 0.01 - 0.1 mg/g DM (Dry Mass),
** 0.1 - 1 mg/g DM,
*** above 1 mg/g DM.
The information about the dry mass ranges represented by asterisks has been added to the footnote of Table 2 and 3.
Page ‘2’, line 343, Table 3: For some values in the MS2 column, decimal places are indicated for some of the values (e.g. No 4, 7,8), but for most of them not. Please be consistent.
This has been corrected. No decimal places are now indicated consistently for any of the positions in the MS2 column.
Page ‘1’, Figure 3 and Page ‘2’, Figure 4: I would recommend using different colors for the PC loading (b) and Salvia abrotanoides in (a), both of which are either the same or a different shade of green. This can lead to confusion.
Thank you for this remark. Green dots representing S. abrotanoides samples on Figure 3a and 4a have been replaced with orange dots to make them clearly distinguishable from green dots representing PC loadings on Figure 3b and 4b and to avoid confusion.
Reviewer 3 Report
Comments:
P 2, line 59 delete word “have”,
p2 line 65 Parallely is not an accepted word. Instead say “In a related study,….”
p5 - lines 227-229
Excess detail on capillary sequencing is given. It is sufficient to note only that the PCR amplicons of ITS and psbA-trnH were sequenced on an ABI 310 instrument.
p5 line 232. Delete word ‘Peaks’ and just note instead that ‘Sequence quality was checked…..”
p5 line 233. I recommend that you revise this sentance as:
Forward and reverse sequencing reads for each marker region were assembled into a contig using BioEdit Sequence Alignment Editor”
p5 line 231. You state you compared the sequences you obtain with data in GenBank. What does it mean that you ‘ compared’ them. I assume you mean to say that you confirmed their identity as ITS and psbA-trnH through similarity to published sequences?
p5 line 242
You state in Line 230 that you generated 8 sequence in total, which reduces down to 2 bi-directional reads of each gene for each of 4 samples. Your phylogeny includes 27 species. So you obtained sequence data for 23 other species from GB. More detail should be added to explain how you decided which sequences in GB to utilize. GB has a lot of incorrect data. Since you use only a single individual for each species, how do we know the taxonomic assignment of the GB sequences is correct? A few words to explain how you verify each sequence from GB would be useful. Lastly - use of an outgroup in your phylogeny would be strongly recommended. Since you mostly use GB sequences, it would be easy enough to add one or two. This will properly orient your molecular phylogeny, even though it will have no effect on the relationship of the two species you sequenced.
pr line 232. When you note that your ‘manually adjusted’ alignment was 540 characters long, what this means is that your alignment cannot be reproduced by another scientist. If you just left the computer generated alignment alone using CLUSTALW, that alignment is fully reproducible. Use of the computer generated alignment is also more objective since manual alignments tend to result in a final alignment which reflects the authors’ bias. Lastly, clustalW is generally thought to perform poorly and methods such as MAFFT are much better, particularly when dealing with gapped alignments. Hard to tell if it would make a difference here, but use of MAFFT would inspire more confidence.
p5 line 236. Note whether you performed alignments for each marker locus separately or whether you concatenated the markers for each individual and then aligned the sequences jointly.
p5 line 241. ML algorithms do not score gaps as part of their model - they are counted as missing data. You state that you employed a Gaps/Missing data treatment. You should explain this a bit more since it is not entirely clear what you did, although you suggest that gaps were an informative character which can alter the phylogeny.
Figure 5. What is 'percentage coverage'? Are these some type of support value, equal to support inferred from bootstrap replicates? This term does not appear to be described in the methods section.
p17 line - Discussion line 439. note: The line number were wrapped through the headers of the figures & tables, page numbers were reset at page 9 when i printed the document.
There are 2 accepted DNA barcode markers, as defined by the willingness of GenBank to assign the keyword “Barcode” for two plant markers: rbcL and matK. Its ok to note that you adopt an ‘extended’ marker framework for DNA barcodes - but your paper does not really fit typical DNA barcode structure due to using only 1 individual for S abrotanoides, and lack of any of the typical metrics like a ‘barcode-gap’ to describe the genetic differences. This is more of a standard molecular phylogeny study than it is a DNA Barcode study. The essence of DNA barcoding is using multiple accessions of each species to evaluate the genetic variance within and among species. You cannot do this with your current data.
page 17 - line 468. You state that psbA-trnH in combination with ITS2 have sufficient resolution to discriminate plant organism. Sounds like you mean that for ALL plants - which would be false. The reason ITS was not chosen as a DNA barcode is in part due to the widespread hybridization among species. If you employed multiple ITS sequences for each species, your phylogeny would not be so clean and neat.
I suggest you keep the statements focused on this section of Salvia.
Line 474. If you included multiple accessions of S abrotanoides in your analysis I might agree that the species can be resolved. When using 1 accession of S abrotanoides, and three of S yangii, I don’t trust that conclusion.
At a minimum, the authors must clearly explain the process by which GenBank data was selected for inclusion.
Author Response
Reviewer 3
Dear Reviewer,
Thank you very much for the detailed revision and corrections proposed to improve our manuscript. We do appreciate the time and effort spend to improve our work. We did our best to suitably address all issues raised by performing appropriate changes into the manuscript and provided extensive explanations in the correspondence below (in blue).
We do hope that you will find this revised manuscript appropriate for publishing in Cells.
Best regards,
Monika Bielecka & Co-authors
Comments:
P 2, line 59 delete word “have”,
The word “have” has been deleted.
p2 line 65 Parallely is not an accepted word. Instead say “In a related study,….”
It has been changed to the “In a related study,…”
p5 - lines 227-229
Excess detail on capillary sequencing is given. It is sufficient to note only that the PCR amplicons of ITS and psbA-trnH were sequenced on an ABI 310 instrument.
Two sentences bringing too much detail on capillary sequencing were deleted (Section 2.5).
p5 line 232. Delete word ‘Peaks’ and just note instead that ‘Sequence quality was checked…..”
It has been changed to the ‘Sequence quality…”.
p5 line 233. I recommend that you revise this sentance as: Forward and reverse sequencing reads for each marker region were assembled into a contig using BioEdit Sequence Alignment Editor”
The previous sentence was deleted and replaced with the above one.
p5 line 231. You state you compared the sequences you obtain with data in GenBank. What does it mean that you ‘ compared’ them. I assume you mean to say that you confirmed their identity as ITS and psbA-trnH through similarity to published sequences?
The previous sentence was deleted and replaced with: “The identity of all obtained sequences was confirmed as ITS or trnH-psbA through similarity to published sequences using the BLAST algorithm.”
You state in Line 230 that you generated 8 sequence in total, which reduces down to 2 bi-directional reads of each gene for each of 4 samples. Your phylogeny includes 27 species. So you obtained sequence data for 23 other species from GB. More detail should be added to explain how you decided which sequences in GB to utilize. GB has a lot of incorrect data. Since you use only a single individual for each species, how do we know the taxonomic assignment of the GB sequences is correct? A few words to explain how you verify each sequence from GB would be useful. Lastly - use of an outgroup in your phylogeny would be strongly recommended. Since you mostly use GB sequences, it would be easy enough to add one or two. This will properly orient your molecular phylogeny, even though it will have no effect on the relationship of the two species you sequenced.
Thank you for this remark. We fully agree with your statement, as, indeed, GB contains a lot of incorrect data. Bearing that in mind, we have been selecting our barcode sequence dataset carefully, avoiding unverified sources, which might bring incorrect data. Therefore, sequences obtained with a voucher-registered specimen were our priority, as a reliable source of data. Sixteen out of 25 sequences downloaded from GB have vouchers. Another six has been isolated by Walker and Sytsma, 2007 [12] and used for their comprehensive phylogenetic analysis, thus regarded as reliable. Only Salvia koyamae, Lamium maculatum and Lamium amplexicaule do not have vouchers, the two latter served as an outgroup though. We were hoping we could rely on sequences verified by BOLD database, but its resources appeared to be too limited. Additionally, we have been only selecting barcode sequences with the same voucher, meaning both ITS and trnH-psbA barcodes were obtained from the same plant specimen. To include this information within the manuscript, we have added additional column into the Table 5 indicating specimen voucher or other source of data. We truly hope that making this information available in the manuscript will sufficiently address this issue.
pr line 232. When you note that your ‘manually adjusted’ alignment was 540 characters long, what this means is that your alignment cannot be reproduced by another scientist. If you just left the computer generated alignment alone using CLUSTALW, that alignment is fully reproducible. Use of the computer generated alignment is also more objective since manual alignments tend to result in a final alignment which reflects the authors’ bias. Lastly, clustalW is generally thought to perform poorly and methods such as MAFFT are much better, particularly when dealing with gapped alignments. Hard to tell if it would make a difference here, but use of MAFFT would inspire more confidence.
The statement in the „Materials and methods” part was certainly unclear. By „manually adjusted” we meant connecting sequences of two alignments, previously cured with Gblocks v.091b. We fully agree with your statement that ClustalW is an imperfect algorithm, thus, we always cure the alignments with Gblocks v.091b – to eliminate poorly aligned regions. Following your recommendation, we have also recalculated the phylogram with MAFFT-generated alignment, the differences, however, were insignificant (as you presumed), thus we decided to retain previously used approach (with further improvements, according to your suggestions).
p5 line 236. Note whether you performed alignments for each marker locus separately or whether you concatenated the markers for each individual and then aligned the sequences jointly.
Alignments were performed separately for each marker and then we have combined them into two-marker matrix. Appropriate amendments were done in the manuscript text (Section 2.5).
p5 line 241. ML algorithms do not score gaps as part of their model - they are counted as missing data. You state that you employed a Gaps/Missing data treatment. You should explain this a bit more since it is not entirely clear what you did, although you suggest that gaps were an informative character which can alter the phylogeny.
Indeed, this is a very apt point. The phylogram was calculated according to the guidelines in the „Phylogenetic Trees Made Easy” by Barry G. Hall. The author suggested Maximum Composite Likelihood as a default model and in the case when there is a lot of gaps in the alignment the „Pairwise deletion” is a suggested data treatment. We calculated the tree with two different data treatments – pairwise and complete deletion. However, when utilizing „pairwise deletion” we have obtained higher support values.
Figure 5. What is 'percentage coverage'? Are these some type of support value, equal to support inferred from bootstrap replicates? This term does not appear to be described in the methods section.
In fact, 'percentage coverage' seems to be not equal to the bootstrap support value. To make our phylogram more reliable and easier to interpret we have included the bootstrap method into our calculations which generated support values. Figure 5 presents now a phylogram of genetic relationship analysis by Neighbor-Joining method based on the Maximum Composite Likelihood model and Bootstrap phylogeny test (1000 replicates) with support values instead of 'percentage coverage' indicated above nodes. Figure 5 caption was changed accordingly.
p17 line - Discussion line 439. note: The line number were wrapped through the headers of the figures & tables, page numbers were reset at page 9 when i printed the document.
Inconsistency in page numeration happened due to introducing sections into the manuscript file, which were needed to present large tables (Table 2 and 3) in a landscape orientation. The page numeration has been corrected.
The line number wrapping through headers must have happened due to some editing issues, which we have not encountered. We hope, such issues will be rectified by the MDPI publication editing service.
There are 2 accepted DNA barcode markers, as defined by the willingness of GenBank to assign the keyword “Barcode” for two plant markers: rbcL and matK. Its ok to note that you adopt an ‘extended’ marker framework for DNA barcodes - but your paper does not really fit typical DNA barcode structure due to using only 1 individual for S abrotanoides, and lack of any of the typical metrics like a ‘barcode-gap’ to describe the genetic differences. This is more of a standard molecular phylogeny study than it is a DNA Barcode study. The essence of DNA barcoding is using multiple accessions of each species to evaluate the genetic variance within and among species. You cannot do this with your current data.
To perform the genetic relationship analysis of S. abrotanoides and S.yangii we were following recommendations published by Hollingsworth et al., 2011 “Choosing and using a plant DNA barcode.” Table 1 in the above paper lists eleven DNA regions that have been included in plant barcoding studies. Indeed, as you say, the strongest recommendation is for the combination of rbcL and matK as the core-barcode for land plants. However, if the discrimination success of this combination turns out to be not sufficient, it is recommended to extend the study with additional markers as required. A particular recommendation goes to trnH-psbA and nrITS/nrITS2.
In the Discussion section of the manuscript we admit that, prior to using trnH-psbA+ITS combination, we have performed analysis with rbcL+matK, which unfortunately turned out to possess little (about 1%) genetic diversity within Salvia genera. Therefore, we have extended our analysis with trnH-psbA+ITS, obtaining fairly satisfactory results.
Regarding the number of S. abrotanoides individuals, we would be happy to include more of them, unfortunately, at present, we do not have an access to other taxonomically undoubtful accessions of S. abrotanoides, thus we decided to rely on what is available. We agree with your opinion and understand that including a single individual into the analysis is not allowed for a DNA barcode study, therefore, we have followed your recommendation and included an outgroup to perform a phylogeny study. A new phylogeny tree was generated (Figure 5) and sections 2.5 and 3.3 were changed accordingly to used methodology and obtained results. Conclusions (in the section 4) were updated accordingly (see also communication below). We truly hope that you will find our improved analysis appropriate for publication.
page 17 - line 468. You state that psbA-trnH in combination with ITS2 have sufficient resolution to discriminate plant organism. Sounds like you mean that for ALL plants - which would be false. The reason ITS was not chosen as a DNA barcode is in part due to the widespread hybridization among species. If you employed multiple ITS sequences for each species, your phylogeny would not be so clean and neat.
I suggest you keep the statements focused on this section of Salvia.
Indeed, the sentence you mentioned might have been interpreted as a general conclusion applicable for all plant organisms – which obviously is false and was not intended. We have changed the sentence to: “In our approach of an extended marker framework for DNA barcodes, a combination of trnH-psbA and ITS2 barcodes have shown sufficient discriminatory power to identify analysed plant organisms to the species level.”, which clearly restricts drawing conclusions to our dataset exclusively.
Line 474. If you included multiple accessions of S abrotanoides in your analysis I might agree that the species can be resolved. When using 1 accession of S abrotanoides, and three of S yangii, I don’t trust that conclusion.
We agree with this criticism. We have changed the concluding sentence to a more elaborated form, pointing out the downside of our analysis: “In the present form, our molecular analysis may suggest that S. abrotanoides and S. yangii are very closely related, yet regarded as two separate species. To make this conclusion fully justified, multiple accessions of S. abrotanoides would have to be included into the analysis, with origin from other populations. Unfortunately, at present, we do not have an access to other taxonomically undoubtful accessions of S. abrotanoides (several specimens from a couple of Botanic Gardens in Europe, gave unclear results, most likely suggesting a hybrid origin, therefore excluded from further studies), thus we decided to rely on what was available.”
At a minimum, the authors must clearly explain the process by which GenBank data was selected for inclusion.
We truly hope, we have met this criterium and you find our explanations sufficient.
Round 2
Reviewer 1 Report
My questions were properly answered and the suggestions were properly inserted in the manuscript.
Reviewer 3 Report
I am satisfied with the replies & revisions made by the author and recommend this article for publication.